# Neighbourhood Ethnic Density, Local Language Skills, and Loneliness among Older Migrants—A Population-Based Study on Russian Speakers in Finland

**DOI:** 10.3390/ijerph20021117

**Published:** 2023-01-08

**Authors:** Laura Kemppainen, Teemu Kemppainen, Tineke Fokkema, Sirpa Wrede, Anne Kouvonen

**Affiliations:** 1Faculty of Social Sciences, University of Helsinki, P.O. Box 4 (Yliopistonkatu 3), 00014 Helsinki, Finland; 2Department of Geosciences and Geography, University of Helsinki, P.O. Box 4 (Yliopistonkatu 3), 00014 Helsinki, Finland; 3Netherlands Interdisciplinary Demographic Institute (NIDI)-KNAW/University of Groningen, Lange Houtstraat 19, 2511 CV The Hague, The Netherlands; 4Department of Public Administration and Sociology, Erasmus School of Social and Behavioural Sciences, Erasmus University Rotterdam, 3062 PA Rotterdam, The Netherlands; 5Centre for Public Health, Institute of Clinical Science, Queen’s University Belfast, Block A, Royal Victoria Hospital, BT12 6BA Belfast, Ireland

**Keywords:** loneliness, older migrants, neighbourhood, language proficiency, mixed model

## Abstract

So far, little attention has been paid to contextual factors shaping loneliness and their interaction with individual characteristics. Moreover, the few existing studies have not included older migrants, identified as a group who are vulnerable to loneliness. This study examined the association between neighbourhood ethnic density (the proportion of own-group residents and the proportion of other ethnic residents in an area) and loneliness among older migrants. Furthermore, we investigated whether local language skills moderated this association. A population-based representative survey (The CHARM study, *n* = 1082, 57% men, mean age 63.2 years) and postal code area statistics were used to study Russian-speaking migrants aged 50 or older in Finland. The study design and data are hierarchical, with individuals nested in postcode areas. We accounted for this by estimating corresponding mixed models. We used a linear outcome specification and conducted logistic and ordinal robustness checks. After controlling for covariates, we found that ethnic density variables (measured as the proportion of Russian speakers and the proportion of other foreign speakers) were not associated with loneliness. Our interaction results showed that increased own-group ethnic density was associated with a higher level of loneliness among those with good local language skills but not among those with weaker skills. Good local language skills may indicate a stronger orientation towards the mainstream destination society and living in a neighbourhood with a higher concentration of own-language speakers may feel alienating for those who wish to be more included in mainstream society.

## 1. Introduction

The effect of loneliness on the health and quality of life of older people is a growing social concern. Loneliness is associated with physical and mental health problems, increased health service use, poor life satisfaction, harmful health behaviours, stress, sleep problems, cognitive decline, and premature mortality [1,2,3,4]. Typically, scholars follow the definition of Perlman and Peplau [5,6] who describe loneliness as a subjective negative experience resulting from inadequate meaningful social connections. These inadequate connections may be experienced in relation to a range of domains, such as a significant other, family and friends, community life, collective identity, and the wider society [6]. Older migrants are generally considered as particularly vulnerable to loneliness [1,2,7].

Studies on the determinants of loneliness predominantly focus on the risk factors at the individual level [2]. Despite an increasing awareness that loneliness is not an exclusively individual phenomenon but is embedded in wider contexts, the call to examine the impact of environmental factors and their interaction with individual characteristics [2,8,9,10] has not been adequately addressed. The importance of place in defining identities, living conditions, social networks, mobility, and quality of life increases with age [11]. Becoming older can intensify feelings about locality, space, and the neighbourhood, which may significantly contribute to quality of daily life [12]. However, prior analyses have rarely accounted for migration histories and ‘ageing out-of-place’ [13], i.e., ageing far from familial, cultural, and social landscapes (however, see [14]). In other words, there is a paucity of research on the extent of spatial variation in the level of loneliness among older migrants and the contextual factors that contribute to this. The present study focuses on older migrants and examines the association of loneliness with neighbourhood level factors in general and ethnic density—the proportion of own-group and other ethnic minority residents in an area—in particular. Moreover, we study whether local language skills moderate this association. We investigate Russian-speaking older migrants in Finland, using data from a population-based representative survey and postal code area statistics [15].

A total of almost 87,500 Russian speakers comprise approximately 20% of all foreign language speakers in Finland [16]. Finland was an autonomous part of the Russian Empire from 1809 to 1917 and, up until very recently, there was a lot of trade, cooperation, and day-to-day interaction between Finns and Russians, especially along the border. Compared to many other migrant groups, Russians are culturally closer to the Finnish majority population [17]. However, there is evidence of discrimination against Russian-speaking migrants in Finland [18,19]. In general, most migrants from Russia and the former Soviet Union countries moved to Finland voluntarily and many of them moved for work or marriage [20]. A large number of older migrants with a Russian background consists of the Ingrian Finnish, eligible for residency based on remigration under certain conditions between 1990 and 2011. Most of the Ingrian Finnish return migrants are the descendants of from seventeenth to early twentieth century Finnish migrants to the Ingria region in Russia. During the 20 years of the remigration programme, about 30,000 people moved to Finland; that is approximately half of the total Ingrian Finnish population [20]. The remigration scheme was closed in 2011 and the last remigrates were granted citizenship in 2016. However, the group is very heterogeneous regarding identities, sense of belonging to ‘Finnishness’, spoken languages, and countries of origin [20,21].

### 1.1. General and Migrant-Specific Risk Factors for Loneliness

According to recent reviews [1,2,3,4], common risk factors of loneliness include older age, female gender, poor self-rated health, depression, low income, and living in long-term care or being a caregiver. Loneliness is also associated with living alone, not being married, having limited social networks and low level of social activity, and living far from family [3]. Moreover, loneliness has been shown to result from life changes, such as a house move, shrinking of social network, loss of a spouse, declining health, and loss of driver’s license [1].

An increasing number of studies have documented high loneliness scores among selected groups of older migrants in the Netherlands [22,23,24], Germany [25], and Canada [26,27] and among older ethnic minority populations in the UK [28,29], compared to the respective majority populations. Older migrants are particularly susceptible to loneliness partly because of their vulnerable position regarding general risk factors [25,30]. Older migrants are often in a more disadvantaged position in several domains, including socioeconomic status, health, participation in larger society (including regular welfare and care institutions), and the ability to take control of one’s own life, which can all expose one to loneliness [31,32].

In addition, the higher levels of loneliness among older migrants may be attributed to migrant-specific factors, such as a weak sense of belonging to the destination society, lack of meaningful relations with non-related co-ethnic peers, intergenerational conflicts, lack of support networks, missing one’s country of origin, and experiencing ethnic discrimination [26,30,33,34,35,36,37,38,39,40,41,42,43]. However, low local language proficiency is the most-often cited risk factor in older migrant’s narratives in this respect [33,34,35,44,45]. According to De Jong Gierveld et al. [26], migrants from countries with similar languages and cultural values and norms to the destination society experience less loneliness than migrants from countries with different languages and cultures. Similarly, Surinamese older migrants reported less loneliness in the Netherlands than Moroccans and Turkish older migrants [23]. As a former Dutch colony, Suriname shares historic ties with the Netherlands and the Surinamese speak Dutch, which might explain their lower levels of loneliness. 

### 1.2. Neighbourhood Level Factors in Loneliness

While there is a growing understanding of the higher level of loneliness among older migrants and its causes, insights into the contextual variation in, and the determinants of, older migrants’ loneliness are missing. Prior neighbourhood studies on loneliness among the general population have shown that spatial variation in contextual factors such as the level of urbanity, population density, socioeconomic deprivation, population change and diversity, social cohesion, infrastructure, transportation, and regional remoteness are associated with the experience of loneliness [1,46,47,48,49,50,51,52,53]. People from ethnic minorities tend to settle in certain neighbourhoods due to a variety of reasons, such as discriminatory housing practices, need for security against racial discrimination, harassment, lower housing costs, and wish to reside close to co-ethnics [54,55,56]. Some studies discussing this ‘ethnic density’ emphasise the positive side of residential concentration providing stronger social networks and reciprocity, more interpersonal communication, survival strategies, ethnic entrepreneurship, and maintaining social cohesion and inclusion [57,58,59,60,61,62,63]. However, research on residential segregation often highlights the possibility of negative ‘neighbourhood effects’ that result from the concentration of minority groups in the urban space; the rationale is that the residents of segregated neighbourhoods lack resources, social connections outside their own ethnic group, and job opportunities that limit their chances of escaping poverty and isolation [64,65].

To our best knowledge, the only study that has examined the role of ethnic density on migrants’ loneliness is by Tseng et al. [62], which found that working-age Chinese migrants living in areas of higher own-group density in Philadelphia reported more family support and lower levels of loneliness than those living in areas with lower ethnic density. There are no studies studying the role of ethnic density in the health or quality of life of older migrants (for review, see [54]). Many previous studies in younger migrants have not found an association between ethnic density and health or mortality [54,56]. However, in studies where an association has been found, protective effects of ethnic density have been reported more frequently than adverse effects [54,56]. Moreover, the associations of mental health and ethnic density have been shown to vary across different migrant groups [54,62]. Bosqui et al. [54] claim that the evidence on ethnic density effect is mixed mainly due to the measurement of ethnic density as overall ethnic density instead of as own-group ethnic density. Furthermore, the evidence on mechanisms behind the ethnic density effect are still scarce [56,62]. Increased social support, fewer experiences of discrimination, and own-language use in ethnically dense areas have been suggested as protective mechanisms for mental health [54,56,57,66]. Even if ethnically dense areas are often also considered as deprived, the area-level deprivation did not explain mental health differences in the studies reviewed by Bosqui et al. [54]. However, most studies did not adequately control for area deprivation [56]. In general, the evidence on a protective ethnic density effect is stronger in studies analysing own-group density and, thus, Bosqui et al. conclude that it cannot be assumed that living in high overall ethnic density areas is protective for every ethnic minority group (see also [67]). Therefore, they call for more studies on individual ethnic groups. In a similar vein, Pemberton and Phillimore [68] argue that ‘superdiverse’ neighbourhoods characterised by diversity, difference, and/or newness create a space for migrants with ‘visible’ differences to blend in. However, neighbourhood identity based on diversity may alienate less visible migrants and culminate in a new form of ‘white flight’, in which white migrants move out from areas where the majority of residents are not white [68]. Similarly, Shell et al. [57] argue that for those living in ethnically dense areas, fluency in local languages may create more pressure for social mobility and moving out of the area. However, they suggest that living in an area of high own-group ethnic density may buffer against these acculturation stressors by reinforcing a sense of identity and ethnic pride.

In addition to differences between migrant groups, we should also pay attention to differences within ethnic groups, which can moderate the association of ethnic density and loneliness. One important factor regarding migrants’ well-being is local language proficiency, which has been shown to be associated with better mental health [69], and lower levels of loneliness [33,34,35,44,45]. Increased possibilities to use one’s own language in everyday life may explain why high ethnic density could be beneficial for mental health [66]. However, in terms of depressive symptoms, Shell et al. [57] showed that those migrants who were fluent in local languages benefitted more from living in an area of high own-group ethnic density than those who did not speak the local language. The only existing study on ethnic density and loneliness by Tseng et al. [62] controlled for the acculturation score including local language proficiency but they did not report full results tables nor discussed the association of acculturation or local language proficiency with loneliness.

### 1.3. The Present Study

All in all, the prior evidence on neighbourhood determinants of loneliness is scarce and the results are inconsistent. Moreover, there is a paucity of representative studies on the relationship between ethnic density and loneliness, particularly among older migrants in Europe. Our first research question aims to address these gaps:(RQ1) How is ethnic density in the neighbourhood—measured as the proportion of Russian speakers and the proportion of other foreign speakers—associated with loneliness among older Russian-speaking migrants in Finland? 

We measured ethnic density with the first language that is registered in the Finnish population information system. The first language is registered by individuals themselves or by their parents in the case of children [70]. The first language and country of birth are the only indicators of ethnicity included in the population register in Finland and only one first language can be registered. The idea to indicate both own linguistic group and other foreign language group densities stems from Bosqui et al. [54], who emphasise the importance of differentiating between overall ethnic density and own-group ethnic density. 

Furthermore, prior research has found that language proficiency plays an important role in migrants’ mental health [69] and in older migrants’ experiences of loneliness [33,34,35,44,45]. The association of ethnic density with mental health has been found to depend on local language proficiency [57,66]. However, there are no previous studies examining the possible moderating role of local language proficiency on the association of ethnic density and loneliness. Thus, our second research question is:(RQ2) Does the association between ethnic density—measured as the proportion of Russian speakers and the proportion of other foreign speakers—and loneliness depend on one’s local language skills among older Russian-speaking migrants in Finland? 


## 2. Materials and Methods

### 2.1. Data

The Care, Health and Ageing of the Russian-Speaking Minority in Finland (CHARM) study focuses on Russian-speaking residents in Finland who are 50 or older. It explores issues related to health, well-being, public services, access to different forms of care, and the use of digital technology. The data were collected in 2019 via a postal survey with online response options. Finland’s Population information system, which contains all registered residents, was used to select a random sample of 3000 people, who had registered Russian as their first language. The response rate was 36%. The sex-based sample stratification was accounted for using survey weighting. Furthermore, the non-response bias was corrected using a response propensity model based on auxiliary gross sample information from national registers (sex, age, income, pensions, unemployment, and region). There were 1082 participants in the study (57% men; mean age 63.2 years, standard deviation 8.4 years).

The neighbourhood-level variables analysed in this study were derived from Statistics Finland’s Paavo postal code area statistics database [15] and from a separate data set including the proportion of foreign speakers in general and Russian speakers in each postal code area based on Statistics Finland’s statistics (available by request). The data include the 3036 postal codes of mainland Finland. Paavo is an open database with information on the population, socioeconomic and housing structure, and about buildings, dwellings, workplaces, and what the main activities in that postal code are. In comparison to similar international administrative units (such as ZIP codes in the United States or NUTS regions in the EU) that have been used for research, Finnish postal codes are generally relatively small. The average number of residents inside postal codes is 1802 and the average area is 112 km^2^. Therefore, they are a good proxy for describing neighbourhoods [71,72]. 

### 2.2. Indicators

Our outcome variable on loneliness was one item from the Center for Epidemiologic Studies Depression Scale (CES-D) [73]. Loneliness was indicated with the question: “During the last week, how often did you feel in the following way: I felt lonely” (response options: 1 = Rarely, 2 = Some of the time, 3 = Most of the time, and 4 = All of the time). This is an ordinal measurement, but we modelled it as an interval variable for simplicity. However, we also conducted robustness checks with binary (1 vs. 2/3/4) and ordinal estimations.

Our key independent variables measure Russian-speaker and other foreign-speaker (OFS) density. The Russian-speaker density was indicated by dividing the number of Russian speakers in a postal code area by the corresponding total population count (multiplied by 100 to obtain percentage points). In a similar manner, we used the count of the non-Russian foreign language speakers to obtain the OFS density.

In addition, several covariates were included in the analyses. Age (years) and sex (woman/man) were controlled for. We also included age squared to allow for the possibility of curvilinear association between age and loneliness, which has been indicated by prior studies [74]. Marital status (married or cohabiting/divorced or separated/widow or widower/not married) was recoded to married or cohabiting vs. other categories. 

The main reason for migrating to Finland was originally an 8-item question that was recoded to categories of repatriation (moving as a repatriate-Ingrian or repatriate, who has Finnish roots or relatives with Finnish roots), family reasons (new relationship or marriage in Finland), moving for work or education (one’s own or spouse’s), and other reasons (asylum seeking or other reasons).

The indicator of the highest educational level in the country of origin includes the following categories: (i) basic/no education/missing (ii) vocational training, and (iii) higher education.

The employment status was coded from the question on main activity: working full-time or part-time or being self-employed vs. other categories (unemployed or temporarily laid off, on long-term sick leave, disability pension or rehabilitation, on statutory pension, providing care for relatives, children or other family members, or studying).

The income support receipt during the past 12 months was used to measure poverty. Income support is a means-tested, last-resort financial assistance benefit in Finland [75].

The participants were asked to assess their proficiency in local languages (Finnish and Swedish) on a 4-point scale. The response options “I use Finnish or Swedish language in various ways in different situations” and “I can participate on everyday conversations in Finnish or Swedish” were categorised as having good proficiency. The response options “I can cope with simple everyday situations in Finnish or Swedish” and “I do not speak either language at all” were merged to indicate weaker language skills. The question of Finnish citizenship (yes/no) was included as such. 

The education in Finland was measured with a question including the following categories: basic education, high school, vocational education, and higher education. Since receiving higher education in Finland was rare, the variable was recoded to some education vs. no education in Finland/missing. 

The length of stay in Finland (in years) was coded to six categories (0–4/5–9/10–14/15–19/20–24/25+). The years in the current neighbourhood was measured as a continuous variable.

The neighbourhood level indicators were drawn from the PAAVO database (Statistics Finland) and included:Located in an urban municipality vs. no (dummy);Number of residents;Population density (per square km);Residents aged 50 years and over, % of residents;Deprivation index (MCA indicator: educational level, unemployment rate, proportion of households with low income, and median income).

Finally, we included dummy variables for sub-national regions to control for possible between-regional variation. In other words, we included regional fixed effects.

### 2.3. Statistical Analysis

Our study is based on two-level data, with individuals nested in neighbourhoods (operationalised as postal code areas). We take this into account by estimating corresponding multilevel models. The survey weights that correct for sampling design and non-response bias were used in estimation. We used a linear outcome specification but conducted also logistic and ordinal robustness checks. Bivariate and adjusted models were used for RQ1. For the full adjusted model, we added the necessary interaction terms to approach RQ2. The effect sizes are reported in relation to the standard deviation (SD) of the outcome variable.

Finally, we performed a series of robustness checks:Unweighted estimation;Mixed logistic estimation with a cut-off between “Rarely” (1)/“Some of the time” (2);Mixed ordered logit estimation;Random slope model where language skills have both random and fixed effects.

## 3. Results

### 3.1. Descriptive Statistics

Approximately 38% of the respondents had felt lonely at least some of the time during the last week (Table 1). Around 75% of the participants were married or cohabiting, 50% had a higher education degree from their country of origin, and 38% rated their local language (Finnish or Swedish) skills as good. Approximately 40% were employed or self-employed. Approximately half of the respondents had Finnish citizenship and over half had lived in Finland for more than 10 years. The mean length of stay in their current neighbourhood was 11.5 years. In our data, the mean proportion of Russian-speaking residents in the postal code area was 2.4% and varied from 0.22% to 18.9%. For the other foreign speakers group, the mean was 6.5%, which varied from 0.4% to 34.3%.

The correlation coefficient (Table 2) between the proportions of Russian speakers and other foreign speakers was 0.38. The correlation of Russian-speaker density and neighbourhood deprivation was 0.31. The neighbourhood deprivation did not correlate with the proportion of other foreign speakers. The correlation matrix also showed a positive correlation of 0.49 between the proportion of older residents and neighbourhood deprivation.

### 3.2. Linear Mixed Model Results

To address RQ1, we examined the bivariate and adjusted associations between the two ethnic density variables and loneliness with a linear mixed model specification that accounted for survey weighting. In the unadjusted analysis, both Russian-speaker density and other foreign-speaker (OFS) density showed positive and statistically significant associations with loneliness (Table 3). An increase of five percentage points in Russian-speaker density translates into an increase of 0.186 units in loneliness (28% of standard deviation, SD), which appears as a substantially significant effect size. For OFS density, the corresponding increase in loneliness is 0.045 units (7% of SD), which indicates a more modest effect size.

In the full main-effects model, in which we controlled for all the mentioned covariates, we found that the ethnic density variables lost their statistical significance. In the case of Russian-speaker density, this is due to increased uncertainty in the estimation: the point estimate remained roughly the same as in the bivariate analysis, but the estimated standard error is clearly larger. However, there were no serious problems of multicollinearity: the variance inflation factor (VIF) for Russian-speaker density was 2.62, which is relatively low. 

Regarding our robustness checks, mixed ordered estimation showed a positive and statistically significant main effect for Russian-speaker density in the full model; compared to the bivariate model, there was an increase in the point estimate. An unweighted estimation of the full model resulted in a *p*-value of 0.054 for Russian-speaker density, with no changes in the point estimate from the bivariate analysis. In contrast, with random slope specification, there was a decrease of 20% in the point estimate of Russian-speaker density in the full model, which points towards possible confounding.

Balancing all this, we consider it safest to opt for a conservative stance: since one of the robustness checks captures a decrease in the point estimate in the full model, we conclude that there is not enough evidence to reject the null hypothesis in this case. Thus, Russian-speaker density does not have an independent association with loneliness. 

Regarding the OFS density, the coefficient changed the sign, but the loss of significance was due to increase in standard errors; in this case, the VIF was already larger, at 5.26. The robustness checks replicated this finding. We can thus conclude that the OFS density does not have an independent association with loneliness.

The second research question was approached by adding the corresponding interaction terms to the full main effects model: i.e., the interaction model included the interactions (i) local language skills x Russian-speaker density and (ii) local language skills x OFS density. The interaction (i) showed a clear overall statistical significance (*p* < 0.0005), but interaction (ii) did not have any explanatory power (*p* = 0.364). In other words, one’s local language skills moderated the association between the Russian-speaker density and loneliness but not the relationship between OFS density and loneliness.

Figure 1 presents predictive margins [76] for interaction (i) and shows that higher Russian-speaker density was associated with more frequent experiences of loneliness for migrants with good local language skills. In contrast, there was no association among those with weaker skills. 

The moderation had a substantial effect size: for those who had good local language skills, an increase of five percentage points in Russian-speaker density was associated with an increase of 0.46 units in loneliness (70% of SD).

Regarding the interaction analysis, all the checks essentially replicated the results reported above.

## 4. Discussion

Approximately 38% of the respondents reported feeling lonely at least some of the time during the last week. In line with previous findings on the antecedents of loneliness, our results show that women reported being lonelier and that living with a partner and being employed or self-employed reduced loneliness [1,2,3]. Receiving income support (i.e., poverty) increased the risk of being lonely. An interesting finding is that those who had moved to Finland based on marriage or romantic partnership were more likely to be lonely than those who had moved based on remigration. This finding should be further examined in future research. In contrast to previous studies [33,34,35,44,45], local language skills were not significantly associated with loneliness as such, but they become important in our interaction results. Our descriptive results also brought about new information on the living conditions of Russian-speaking migrants in Finland. The neighbourhood level correlations showed that, relatively, older Russian-speaking migrants often live in deprived neighbourhoods. In general, our respondents lived in urban municipalities and were concentrated in the capital region.

Our first research question asked how own-linguistic group density and other foreign-speaker density in the neighbourhood were associated with loneliness. A previous study in working-age Chinese migrants in Philadelphia showed that migrants living in areas of higher own-group density reported less loneliness and more family support than those living in areas of lower own-group ethnic density [62]. Other studies have reported mixed findings of the protective effect of ethnic density on mental health [54,56]. In the present study, neither own-linguistic group density nor other foreign-speaker density remained associated with loneliness after we controlled for covariates. These contradictory findings reveal the need for more research on the associations between living conditions and loneliness among different migrant groups in different national political and welfare systems.

In addition, other neighbourhood characteristics, such as area deprivation, number of residents, urbanity, or population density, were not associated with individual loneliness. However, there are differences across regions so that participants living in the capital region and Keski-Pohjanmaa (Swedish-speaking area) reported higher levels of loneliness than participants residing in other regions. These regional differences merit more research in the future.

Our second research question examined whether the association between ethnic density and loneliness depended on one’s local language skills. Our interaction results showed that higher own-group ethnic density was associated with higher levels of loneliness among those with good local language skills but not among those with weaker skills. Based on the previous research on loneliness and the mechanisms moderating the associations between ethnic density and mental health, we could assume that higher own-group ethnic density would protect older migrants with weaker local language skills from loneliness [66]. In our data, this was not the case: ethnic density had no role in defining the level of older migrants’ loneliness among those with weaker local language proficiency. However, it should be noted that ethnic density was not associated with higher levels of loneliness either. 

Yet, our results showed that the proportion of own-linguistic group speakers in one’s neighbourhood is an important factor shaping older migrants’ loneliness. Especially, those with good local language skills tend to feel lonelier in areas characterised by high own-linguistic group density. Good local language skills may indicate a stronger orientation towards the mainstream destination society and living in a neighbourhood with a higher concentration of Russian-speaking migrants may feel alienating for those who wish to be more included in the Finnish mainstream. Albert [42] showed in their research among older Portuguese migrants in Luxembourg that migrants who experienced difficulties in reconciling different cultural belongings reported more loneliness than those with compatible identity orientation. This identity conflict may also explain our results; living in an area with many co-ethnics while orienting strongly to the mainstream society may create an identity crisis and a sense of belonging to neither mainstream society nor co-ethnic community. As Klok et al. [43] have found, integration into mainstream society can be as protective as a sense of belonging to one’s own ethnic group for reducing loneliness. The essence here is that, to not to feel lonely, one needs to feel a sense of belonging to a certain group instead of being marginalised. Recent intervention projects across Europe have addressed loneliness but their effects have been limited due to insufficient understanding of the phenomenon [4,9,10]. It is important that these projects acknowledge the intersecting risk factors at both individual and neighbourhood levels.

A key strength of our study is that it is the first one to examine how environmental factors interact with individual factors to shape loneliness among older migrants. Another strength of our study is the use of high-quality representative data of older migrants. Large-scale representative surveys on older migrants are very rare. The main limitations of our study stem from the well-known limitations of cross-sectional observational studies. Most importantly, reverse causality and unobserved confounders limit causal inference. Another limitation of our study is related to generalisability and comparisons of our results to other older migrant populations. Future studies should aim at collecting comparable data among different ethnic groups in the destination country. As resources rarely allow for these types of large comparison studies, a series of case studies could increase our understanding too.

A final limitation is related to the conceptualisation of ethnic group and ethnic density. Implementing the notion of ethnic density in a European migration context demonstrates that the concept requires more careful attention. Ethnic identity formation is a complex and dynamic social process. As, for example, Brubaker [77] notes, it is important for scholars to adopt a critical and self-reflexive stance towards the categories we use, considering whether they are related to self-identification or other-identification. In this vein, we observe that, whereas Russian-speaking migrants in our study constitute a population group, the issue regarding their ethnic identities is an empirical question that cannot be assumed only on the basis of a shared first language. Imaginings about a community based on shared nationality may underpin a diasporic identity of ‘Russianness’, but many Russian speakers may adhere to other national or ethnic identities, perceiving themselves as Ukrainians, people from Belarus, Ingrian Finnish, etc. [78]. Accordingly, the research on ethnic density should be developed with greater sensitivity to migration context and migration histories that help researchers to delineate whether a minority population group is internally diverse when it comes to issues such as ethnic identities, social class, and religion. As a more technical note, it should be observed that ‘ethnic density’ is a rather problematic term, since it may be interpreted to imply the idea that some groups are more ‘ethnic’ than others and that ethnicity is a stable, measurable characteristic of individuals and, accordingly, neighbourhoods. Future studies could further develop this line of thought to find a terminological solution that overcomes such connotations but still retains sufficient analytical power to further examine the implications of neighbourhood ethnic composition and related social relations. Moreover, future research should also incorporate subjective feelings of belonging and see how one’s sense of belonging regarding ethnic or national groups is associated with feelings of loneliness or how an individual sense of belonging moderates the association between ethnic density and loneliness or mental health, for example.

## 5. Conclusions

Previous studies have not adequately explained how environmental and individual factors interact to shape loneliness. Moreover, this question has not previously been investigated among older migrants. Our study examined how neighbourhood characteristics, particularly ethnic density, is associated with loneliness among older Russian-speaking migrants in Finland. Our results showed that the role of ethnic density is not straightforward but depends on the command of local language: higher own-linguistic group density was associated with a higher level of loneliness among those with good local language skills but not among those with weaker local language skills. The people living in areas with high own-linguistic group density may feel alienated due to an identity conflict between their own ethnic group and the mainstream society. To reduce loneliness and increase migrant well-being, scholars and policymakers should pay careful attention to within-group differences and be aware that loneliness can be influenced by several intersecting factors both at the individual and neighbourhood levels.

## Figures and Tables

**Figure 1 ijerph-20-01117-f001:**
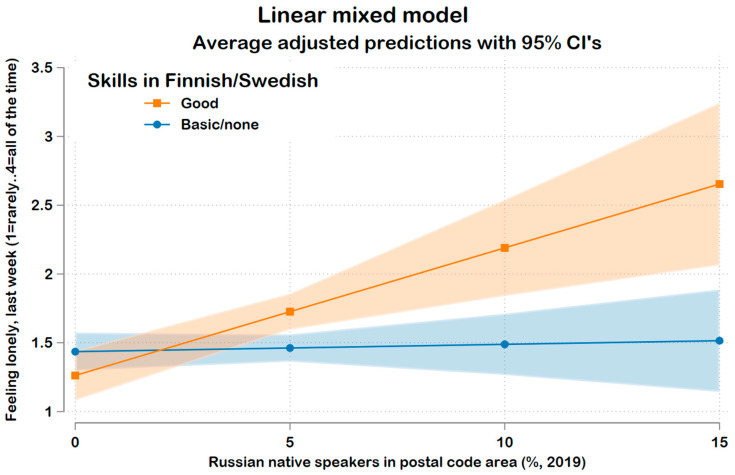
The association between Russian-speaker density and loneliness is moderated by local language skills (*p* < 0.0005).

**Table 1 ijerph-20-01117-t001:** Descriptive statistics.

Individual Level Variables		n	%	Obs.
Loneliness	Rarely	632	62.5	1012
	Some of the time	317	31.3	1012
	Most of the time	47	4.6	1012
	All of the time	16	1.6	1012
Sex *	Women	466	43.1	1082
Married or cohabiting	Yes	796	74.6	1067
Main reason of migration	Repatriation	552	52.0	1061
	Family	258	24.3	1061
	Work or education (own/spouse)	194	18.3	1061
	Other	57	5.4	1061
Highest education in country of origin	Higher education	541	50.0	1082
	Vocational education	459	42.4	1082
	Basic/no education/missing	82	7.6	1082
Employed or self-employed	Yes	426	39.4	1082
Recipient of income support	Yes	421	41.6	1013
Local language skills	Basic/none	634	62.2	1019
	Good	385	37.8	1019
Finnish citizenship	Yes	525	49.0	1071
Education in Finland	No education/missing	682	63.0	1082
	Some education	400	37.0	1082
Years in Finland	0–4	89	8.4	1056
	5–9	158	15.0	1056
	10–14	166	15.7	1056
	15–19	176	16.7	1056
	20–24	224	21.2	1056
	25+	243	23.0	1056
		**Mean (std.dev)**	**Min/max**	**Obs.**
Age *		63.2 (8.4)	50/93	1082
Years in current neighbourhood		11.5 (8.3)	0/51	1052
**Area Level Variables**		**n**	**%**	**Obs.**
Type of municipality	Urban	347	75.8	458
	Semi-urban/rural	111	24.2	458
Region	Etelä-Karjala	34	7.4	458
	Etelä-Pohjanmaa	5	1.1	458
	Etelä-Savo	20	4.4	458
	Kainuu	7	1.5	458
	Kanta-Häme	14	3.1	458
	Keski-Pohjanmaa	2	0.4	458
	Keski-Suomi	21	4.6	458
	Kymenlaakso	40	8.7	458
	Lappi	8	1.8	458
	Pirkanmaa	26	5.7	458
	Pohjanmaa	6	1.3	458
	Pohjois-Karjala	23	5.0	458
	Pohjois-Pohjanmaa	17	3.7	458
	Pohjois-Savo	26	5.7	458
	Päijät-Häme	34	7.4	458
	Satakunta	16	3.5	458
	Uusimaa	134	29.3	458
	Varsinais-Suomi	25	5.5	458
		**Mean (std.dev)**	**Min/max**	**Obs.**
Russian-speaker density		2.38 (2.1)	0.2/18.9	412
Other foreign-speaker density		6.48 (6.5)	0.4/34.3	412
Number of residents		5572.9 (4642.5)	37/27,628	458
Population density (per square km)		1123.0 (1793.6)	0.4/17,246	458
Residents 50 aged years and over, % of residents		0.4 (0.1)	0.2/0.8	458
Deprivation index (MCA indicator)		0.0 (1.1)	−3.8/3.4	455

* Sex and age are from the population register; other indicators at the individual level are from the survey.

**Table 2 ijerph-20-01117-t002:** Correlation matrix of the neighbourhood variables.

	1	2	3	4	5	6
(1) Russian-speaker density	1					
(2) Other foreign-speaker density	0.38	1				
(3) Number of residents	−0.04	0.45	1			
(4) Population density (per square km)	0.10	0.48	0.47	1		
(5) Residents 50 years and over, % of residents	0.01	−0.45	−0.41	−0.39	1	
(6) Deprivation index (MCA indicator)	0.31	−0.01	−0.12	−0.24	0.49	1

**Table 3 ijerph-20-01117-t003:** Linear mixed model on loneliness.

	(1) Bivariate Models		(2) Full Main Effects Model		(3) Interaction Model	
	b	*p*	b	*p*	b	*p*
Russian-speaker density	**0.04**	**0.027**	0.04	0.064	0.01	0.740
Other foreign-speaker density	**0.01**	**0.003**	−0.01	0.171	−0.01	0.304
Number of residents	.	.	0.00	0.591	0.00	0.535
Population density (per square km)	.	.	0.00	0.559	0.00	0.501
Residents 50 years and over, % of residents	.	.	−0.35	0.415	−0.18	0.676
Deprivation index (MCA indicator)	.	.	0.06	0.160	0.05	0.237
Urban municipality	.	.	0.03	0.818	0.04	0.735
Region (reference: Uusimaa)						
*Etelä-Karjala*	.	.	**−0.43**	**0.007**	**−0.44**	**0.003**
*Etelä-Pohjanmaa*	.	.	**−0.51**	**0.001**	**−0.48**	**0.007**
*Etelä-Savo*	.	.	−0.29	0.128	−0.33	0.086
*Kainuu*	.	.	−0.05	0.864	−0.08	0.812
*Kanta-Häme*	.	.	**−0.35**	**0.020**	**−0.29**	0.056
*Keski-Pohjanmaa*	.	.	0.40	0.050	**0.55**	**0.005**
*Keski-Suomi*	.	.	**−0.43**	**0.011**	**−0.38**	**0.021**
*Kymenlaakso*	.	.	**−0.41**	**0.004**	**−0.41**	**0.004**
*Lappi*	.	.	−0.08	0.694	−0.11	0.597
*Pirkanmaa*	.	.	−0.16	0.441	−0.13	0.531
*Pohjanmaa*	.	.	−0.20	0.273	−0.20	0.320
*Pohjois-Karjala*	.	.	**−0.62**	**0.000**	**−0.62**	**0.000**
*Pohjois-Pohjanmaa*	.	.	−0.09	0.659	−0.12	0.541
*Pohjois-Savo*	.	.	**−0.44**	**0.015**	**−0.44**	**0.013**
*Päijät-Häme*	.	.	**−0.28**	**0.045**	**−0.26**	**0.048**
*Satakunta*	.	.	**−0.54**	**0.001**	**−0.55**	**0.000**
*Varsinais-Suomi*	.	.	−0.27	0.055	−0.28	0.051
Age	.	.	−0.05	0.250	−0.04	0.311
Age squared	.	.	0.00	0.250	0.00	0.299
Woman	.	.	**0.12**	**0.028**	**0.13**	**0.016**
Married or cohabiting	.	.	**−0.28**	**0.000**	**−0.29**	**0.000**
Main reason of migration (ref: Repatriation)	.	.	.	.	.	.
*Family*	.	.	**0.18**	**0.026**	**0.19**	**0.017**
*Work or education (own/spouse)*	.	.	0.01	0.861	0.01	0.864
*Other*	.	.	0.17	0.114	0.15	0.146
Highest education in country of origin (ref: Higher)						
*Vocational*	.	.	−0.07	0.235	−0.06	0.289
*Basic/no education/missing*	.	.	−0.05	0.648	−0.07	0.467
Employed or self-employed	.	.	**−0.17**	**0.022**	**−0.16**	**0.031**
Recipient of income support	.	.	**0.13**	**0.035**	0.12	0.053
Good local language skills	.	.	0.15	0.069	−0.11	0.263
Finnish citizenship	.	.	−0.04	0.511	−0.04	0.484
Education in Finland	.	.	0.03	0.579	0.02	0.702
Years in Finland (ref: 0–4)						
5–9	.	.	0.08	0.450	0.08	0.436
10–14	.	.	0.11	0.274	0.12	0.246
15–19	.	.	0.16	0.140	0.15	0.167
20–24	.	.	0.00	0.986	−0.01	0.951
25+	.	.	0.17	0.128	0.17	0.108
Years in current neighbourhood	.		0.00	0.739	0.00	0.714
Good local language skills × Russian-speaker density (%, postcode area)	.				**0.09**	**0.000**
Good local language skills × Other foreign-speaker density (%, postcode area)	.				−0.01	0.364
Constant	.		**3.25**	**0.016**	**3.02**	**0.019**
Observations	967		821		821	.
Number of groups	397	.	371	.	371	.
Between-neighbourhoods variance	.	.	0.068	.	0.065	.
Residual variance	.	.	0.339	.	0.331	.

Note: Statistically significant coefficients in bold (*p* < 0.05).

## Data Availability

The data presented in this study are available on request from the corresponding author. The data are not publicly available due to general data protection guidelines.

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
