# Peer review of "Neighbourhood Ethnic Density, Local Language Skills, and Loneliness among Older Migrants—A Population-Based Study on Russian Speakers in Finland"

_ijerph, 2023, doi:10.3390/ijerph20021117_

Round 1
Reviewer 1 Report
Thank you very much for letting me review the present manuscript. This manuscript tackles the interesting question of how ethnic density is related to feelings of loneliness of older Russian-speaking migrants living in Finland, thereby taking into account a number of background variables such as local language competence. I have enjoyed reading this study.
I have several suggestions for clarification though.
On p. 2, line 92, the authors introduce the term "cultures", I would however suggest to give some further explanation about how they conceptualize culture
On p. 4, line 175 the authors state: "The idea to indicate both own linguistic group and other foreign language group densities stems from Bosqui et al.", however I don't understand why this remark is given here, I think it is not clear at this point why this sentence is relevant (it becomes clear afterwards when the operationalization of ethnic density is explained).
On p. 4, the authors state that "However, Shell et al. [49] showed that those migrants who were fluent in local languages benefitted more from living in an area of high own-group ethnic density in terms of mental health than those who did not speak the local language." but later on that page in the research questions they state that "However, there are no previous studies examining the possible moderating role of local language proficiency on the association of ethnic density and loneliness." Considering the mentioned study by Shell et al., this second statement would not be correct, would it?
On p. 5, line 203: please correct typos (consisted of; the group is very heterogeneous): "to the Ingria region in Russia. During the 20 years of the remigration programme, some 30,000 people moved to Finland, which consisted approximately half of the Ingrian Finnish population [67]. The remigration scheme was closed in 2011 and the last remigrates were granted citizenship in 2016. However, the groups is very heterogenous in regard to identities, sense of belonging to ‘Finnishness’, spoken languages and countries of origin [67,68]
On p. 5, line 245: why was age squared used?
I was wondering how the main reason for migration was coded? There are more than two categories, thus it cannot be simple dummy coding.
On p. 6: line 255: I was wondering what "Neighbourhood level" means
On p. 6, line 264: "Finally, the larger sub-national region was controlled for (regional fixed effects).=> at this point, it's not clear
On line 276: random slope model where language skills have both random and fixed effect3.=> what is effect3?
With regard to table 1: the "Highest education in the country of origin" is mentioned but I was wondering what if they studied abroad?
With regard to the first analysis, I wonder why Region (reference: Uusimaa) was chosen as reference. If this is the region of most interest, wouldn't it be easier to create a simple dummy with Uusimaa and other as two categories? Couldn't so many predictors be problematic, statistically speaking?
I wonder if participants who felt more lonely in Russian speaking areas might have had lower Russian language skills - was this assessed too, or was only local language skills assessed?
p. 11, 356: insert comma: "Women reported being lonelier, and living with a partner and being employed or self-employed reduced loneliness."
In the conclusions, I was expecting a discussion regarding the results by Shell et al. which seem to contradict the results in the present study.
Author Response
We thank the reviewer for taking the time to read our manuscript and for their constructive feedback.
Below you can find our point-by-point responses to reviewer’s comments.
On p. 2, line 92, the authors introduce the term "cultures", I would however suggest to give some further explanation about how they conceptualize culture
RESPONSE: thank you for this, we have now clarified the sentence and refer to cultural values and norms, which the cited authors discuss (p.3, line 110-111).
On p. 4, line 175 the authors state: "The idea to indicate both own linguistic group and other foreign language group densities stems from Bosqui et al.", however I don't understand why this remark is given here, I think it is not clear at this point why this sentence is relevant (it becomes clear afterwards when the operationalization of ethnic density is explained).
RESPONSE: We have clarified that we refer to Bosqui et al.’s idea on the importance of differentiating between overall ethnic density and own-group ethnic density, which was explained on page 3. This idea is built into our research question 1, and thus, we think it is important to note here. We now repeat the explanation here too (p.4, line 193-195).
On p. 4, the authors state that "However, Shell et al. [49] showed that those migrants who were fluent in local languages benefitted more from living in an area of high own-group ethnic density in terms of mental health than those who did not speak the local language." but later on that page in the research questions they state that "However, there are no previous studies examining the possible moderating role of local language proficiency on the association of ethnic density and loneliness." Considering the mentioned study by Shell et al., this second statement would not be correct, would it?
RESPONSE: Shell et al. have studied depressive symptoms, not loneliness. We have clarified this in the text (p.4, line 173).
On p. 5, line 203: please correct typos (consisted of; the group is very heterogeneous): "to the Ingria region in Russia. During the 20 years of the remigration programme, some 30,000 people moved to Finland, which consisted approximately half of the Ingrian Finnish population [67]. The remigration scheme was closed in 2011 and the last remigrates were granted citizenship in 2016. However, the groups is very heterogenous in regard to identities, sense of belonging to ‘Finnishness’, spoken languages and countries of origin [67,68]
RESPONSE: thank you, we have corrected the typos (p.2, line 81-86).
On p. 5, line 245: why was age squared used?
RESPONSE: Thank you. We opted for this solution to allow for the possibility of curvilinear association between age and loneliness, which has been indicated by prior studies. We also added this explanation to the text (p.5, line 244-248). It should be noted that this is a safe choice also in the case when there is a linear association: in that case, age squared does not receive a statistically significant coefficient. With relatively large data, the implied cost of one degree of freedom is negligible.
I was wondering how the main reason for migration was coded? There are more than two categories, thus it cannot be simple dummy coding.
RESPONSE: Thank you, we have now rewritten the indicators section, and explained the coding of original variables in more detail (re main reason for migration, see p.5, line 249-p.6, line 253).
On p. 6: line 255: I was wondering what "Neighbourhood level" means
RESPONSE: “Individual level” and “neighbourhood level” were headings. We have now rewritten the part so it should be clearer.
On p. 6, line 264: "Finally, the larger sub-national region was controlled for (regional fixed effects).=> at this point, it's not clear
RESPONSE: This refers to a standard fixed effects approach. For example, in many cross-national studies one finds the case where all between-country differences have been controlled for. In this study we were able to do the same but at a sub-national level. Our formulation has been amended as follows:
“Finally, we included dummy variables for sub-national regions to control for all possible between-regional variation. In other words, we included regional fixed effects.” (p.6, line 287-288)
On line 276: random slope model where language skills have both random and fixed effect3.=> what is effect3?
We are sorry, this was a typo, we have fixed it (effects, p.6, line 302).
With regard to table 1: the "Highest education in the country of origin" is mentioned but I was wondering what if they studied abroad?
RESPONSE: Unfortunately we only asked for the respondents’ education in their country of origin and in Finland.
With regard to the first analysis, I wonder why Region (reference: Uusimaa) was chosen as reference. If this is the region of most interest, wouldn't it be easier to create a simple dummy with Uusimaa and other as two categories? Couldn't so many predictors be problematic, statistically speaking?
RESPONSE: We wanted to control for all between-regions effects to obtain the best possible individual and neighbourhood level estimates. Furthermore, the signs differ, which means that a simple dichotomy would fail analytically. Degrees of freedom pose no problem to this solution. Neither were there serious problems of multicollinearity. We would like to highlight that fixed effects solution is a very standard and established practise. Omitting it would risk inferences based on more biased estimates.
I wonder if participants who felt more lonely in Russian speaking areas might have had lower Russian language skills - was this assessed too, or was only local language skills assessed?
RESPONSE: All the respondents have registered Russian-language as their first language so we can assume that they are fluent in Russian. We have added this information to the data description (p.5, line 213).
- 11, 356: insert comma: "Women reported being lonelier, and living with a partner and being employed or self-employed reduced loneliness."
RESPONSE: Thank you, we have corrected this (p.11, line 380).
In the conclusions, I was expecting a discussion regarding the results by Shell et al. which seem to contradict the results in the present study.
RESPONSE: Our study is the closest in nature to Tseng et al., whose results contradict ours. We have now added discussion on this (p.12, line 393-402).
Reviewer 2 Report
Dear authors,
I enjoyed the opportunity to review your manuscript. I thought the paper was concise. I have a few suggestions for your consideration.
Introduction- Part of what you have in the study context line 187-205 move to the introduction and then under study context describe the study design.
Study design is not clear.
Analytical strategy- line 263- this is confusing. can you call this statistical analysis if that is the case?
Data- in regards to non-response bias- what was the significance of adjusting the responses? Was this the approach of handling the missing responses? if not make it clear how you went about it.
How were the indicators originally measure?
Conclusion/discussion- separate these two. Have discussion by itself and then conclusion.
Discussion- the authors are discussing "earlier studies", "previous studies" but failed to include which those studies are.
Strengths and limitation- This study failed to include strengths and limitations. where there none? I suggest including it in this sections.
Implications- Are there implications/applications of this study? It would be important to include this as well in the conclusion.
Author Response
We thank the reviewer for taking the time to read our manuscript and for their constructive feedback.
Below you can find our point-by-point responses to reviewer’s comments.
Introduction- Part of what you have in the study context line 187-205 move to the introduction and then under study context describe the study design.
RESPONSE: Thank you, we have moved the suggested paragraph to the introduction (p.2, line 69-86).
Study design is not clear.
RESPONSE: We have clarified the description of our statistical analysis (p.6, line 290-292).
Analytical strategy- line 263- this is confusing. can you call this statistical analysis if that is the case?
RESPONSE: Thank you. We have changed the heading as suggested to “Statistical analysis” (p.6, line 289)
Data- in regards to non-response bias- what was the significance of adjusting the responses? Was this the approach of handling the missing responses? if not make it clear how you went about it.
RESPONSE: Thank you very much for this comment. We agree that our description of data weighting was not detailed enough. We have added a new paragraph:
“Sex-based sample stratification was accounted for by survey weighting. Furthermore, non-response bias was corrected by response propensity model based on auxiliary gross sample information from national registers (sex, age, income, pensions, unemployment, region).” (p.5, line 214-216)
How were the indicators originally measure?
RESPONSE: Thank you for the comment. We have now rewritten the indicators paragraph (p.5+6) and believe it is now clearer and easier to read.
Conclusion/discussion- separate these two. Have discussion by itself and then conclusion.
RESPONSE: We have now written a separate conclusion part (p.13, line 476-489).
Discussion- the authors are discussing "earlier studies", "previous studies" but failed to include which those studies are.
RESPONSE: We have added references to the discussion part (p.11, line 381+385).
Strengths and limitation- This study failed to include strengths and limitations. where there none? I suggest including it in this sections.
RESPONSE: Thank you for pointing this out. We have added a section on strengths and limitations in the discussion part (p.12, line 440-p.13, line 451).
Implications- Are there implications/applications of this study? It would be important to include this as well in the conclusion.
RESPONSE: We added discussion on policy implications in the conclusion (p.13, line 486-489).